# Image steganography without embedding by carrier secret information for secure communication in networks

**Yangwen Zhang**[1,2], **Yuling Chen** [ID][1,2] *, **Hui Dou**[1,2], **Chaoyue Tan**[1], **Yun Luo**[1], **Haiwei Sang**[3]

**1** State Key Laboratory of Public Big Data, Guizhou University, Guiyang, China, **2** College of Computer Science and Technology, Guizhou University, Guiyang, China, **3** Guizhou Education University, Guiyang, China

* ylchen3@gzu.edu.cn

**Data Availability Statement:** All relevant data are within the manuscript and its Supporting information files.

**Funding:** This research was supported by Foundation of National Natural Science Foundation

## Abstract

Steganography, the use of algorithms to embed secret information in a carrier image, is widely used in the field of information transmission, but steganalysis tools built using traditional steganographic algorithms can easily identify them. Steganography without embedding (SWE) can effectively resist detection by steganography analysis tools by mapping noise onto secret information and generating secret images from secret noise. However, most SWE still have problems with the small capacity of steganographic data and the difficulty of extracting the data. Based on the above problems, this paper proposes image steganography without embedding carrier secret information. The objective of this approach is to enhance the capacity of secret information and the accuracy of secret information extraction for the purpose of improving the performance of security network communication. The proposed technique exploits the carrier characteristics to generate the carrier secret tensor, which improves the accuracy of information extraction while ensuring the accuracy of secret information extraction. Furthermore, the Wasserstein distance is employed as a constraint for the discriminator, and weight clipping is introduced to enhance the secret information capacity and extraction accuracy. Experimental results show that the proposed method can improve the data extraction accuracy by 10.03% at the capacity of 2304 bits, which verifies the effectiveness and universality of the method. The research presented here introduces a new intelligent information steganography secure communication model for secure communication in networks, which can improve the information capacity and extraction accuracy of image steganography without embedding.

## 1 Introduction

Information hiding techniques include steganography [1, 2] and watermarking [3, 4]. Digital watermarking is the process of embedding a watermark into a cover image by means of an algorithm or a deep neural network, which is used to protect the copyright of an image without the human eye being able to detect the watermark. Aditya [4] proposes a logistic map based fragile watermarking technique, which takes advantage of the sensitivity property of the

of China (62202118), and Scientific and Technological Research Projects from Guizhou Education Department (Qian jiao ji [2023]003), and Provincial Department of Science and Technology's Hundred level Innovation Talents Project (Guizhou Science and Technology Cooperation Platform Talents-GCC [2023] 018), Guizhou Province Major Project (Qiankehe Major Project No. [2024] 003), and Top Technology Talent Project from Guizhou Education Department (Qian jiao ji [2022]073). (By Corresponding authors: Yuling Chen). Guizhou Provincial Basic Research Program(Natural Science):ZK[2024] (652) and Science and Technology Program of GuiYang:(ZK[2024]-1-2). (by Authors: Haiwei Sang) The funders had no role in study design, data collection and analysis, decision to publish, or preparation of the manuscript.

**Competing interests:** The authors have declared that no competing interests exist.

logistic map to generate the watermark bits. Steganography is the process of hiding secret messages within a medium to create a steganographic carrier, which is then used to secure communication in networks [5], where steganography using images as the carrier medium is known as image steganography.

In the field of steganography without embedding, steganography without embedding is able to effectively resist the recognition of steganography analysis tools due to the fact that it does not directly employ the embedding of secret message into the cover image, thus achieving the purpose of securely transmitting information. The field of image steganography without embedding has attracted considerable interest among researchers due to its characteristics. However, the majority of studies on image steganography without embedding have been unable to effectively enhance the capacity and extraction accuracy of secret message while maintaining security. Previous methods have been to have limitations in terms of image distortion and the accuracy of message extraction, particularly when dealing with large capacities of steganographic message. This has led to a reduction in the usability of image steganography without embedding. This paper examines the potential of image steganography without embedding, with the objective of enhancing the capacity and extraction accuracy of secret message while maintaining a certain level of security.

Traditional image steganography is mainly based on algorithms for embedding of secret information into the carrier image, which is the main part of image steganography. The embedding method is divided into the spatial domain and frequency domain. Representatives of traditional steganography in the spatial domain are the LSB [6, 7], SUNIWARD [8]. Samar et al. [9] proposed a data hiding method that exploits the low embedding capacity and high variability properties of block-wise histogram shifting, thereby enhancing the robustness and embedding capacity of steganography. However, the spatial domain image steganography method will cause the pixel to change value in the steganographic image after the steganographic image is subjected to operations such as cropping, rotating, and scaling, and the steganographic information of the steganographic image will be lost. Traditional frequency domain image steganography hides the steganography in the frequency domain of the carrier image, which significantly reduces the secret information caused by image alteration. Representatives of traditional image steganography in frequency domain space include the discrete cosine transform (DCT) algorithm [10] and so on. Frequency domain steganography algorithms have better robustness compared to spatial domain steganography algorithms and can effectively resist robustness attacks. However, with the advent of steganalysis tools, as proposed by J. Hemalatha et al. [11] proposed the third order subtractive pixel adjacency matrix features with an ensemble classifier steganalysis method, which is able to effectively differentiate the distribution of steganographic images from clean images.

To increase the steganographic capability of the cover image, in recent years, more and more researchers have devoted themselves to steganography with deep networks to solve the problem of satisfying the imperceptibility and accuracy of extracting secret information. Therefore, a model of steganography with deep networks, such as SGAN [12], SSGAN [13], StarGAN [14], is proposed. Deep networks are used to learn the best position of a steganographic image in the base frame, and the steganographic method is applied to convert the hidden information. The information is hidden in the carrier image to hide the secret message inside the carrier image [15]. Nevertheless, the above is easily detected by steganalysis tools, which leads to the failure of transmission of the steganographic image, because the above methods hide the concealed messages by altering the pixels in the carrier image. Based on the above problems, a deep learning-based without embedding steganography method SWE [16] was developed. The steganography process of the SWE method is shown in Fig 1.

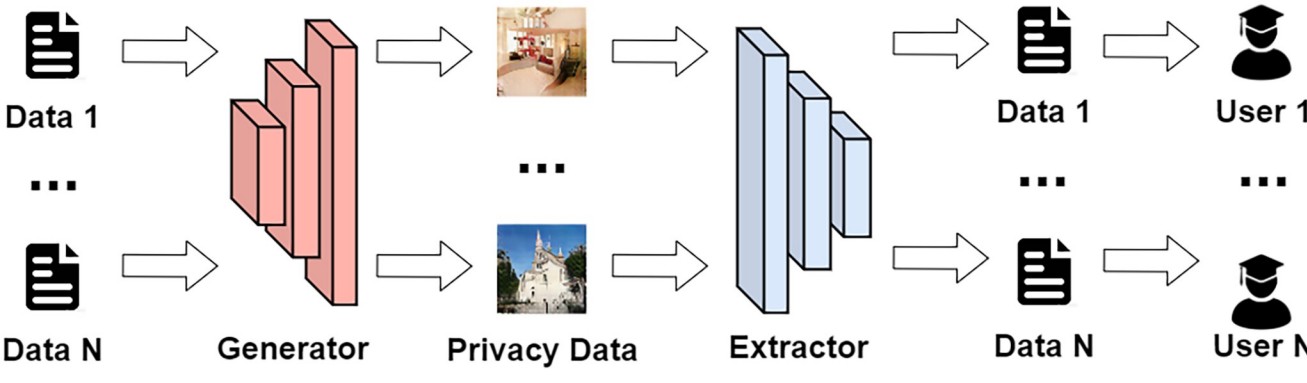

**Fig 1.**

The SWE method maps the secret messages onto the noise carrier of the generated image, instead of generating the steganographic image directly from the hidden information or embedding the secret information into the carrier image. Since the secret information is not directly involved in the steganographic process, the SWE method fundamentally solves the steganalysis problem of typical steganalysis tools. At present, SWE methods are mainly classified into two categories. The first strategy is to generate the corresponding hash sequences [17, 18] through a mapping association mechanism that involves the secret message and the set of available images. Liu et al. [19] proposed the use of DenseNet and DWT features to generate robust feature sequences, with the establishment of an inverted index for secret message and feature sequences. However, the secret message payload of this method is relatively small, and for large data steganography, it requires a substantial existing dataset as the steganographic data carrier, which makes it challenging to be applied on a large scale at present. Another method is to generate synthetic images by training GANs [20] network. GANs network generates images that require random noise and replaces the random noise with secret noise generated by secret message mapping [21]. The generator of GANs network generates steganography by secret noise image. This method has achieved good results but still faces many problems, so the steganography method which is deep learning network generation cannot be used. For example: (1) The quality of stego image generated by steganographic method using GANs to generate images based on deep learning can not reach the level of cover images, which affects the requirements of imperceptibility; (2) The steganographic image generated on the basis of GANs deep learning the payload of this method is not high; (3) When training the extractor to extract information from the steganographic image, there are difficulties in extracting the steganographic information stably.

The majority of existing SWE methods are afflicted by a number of shortcomings. These include the quality of steganographic images, which is often insufficient to achieve the desired effect of cover images; the capacity of steganographic information in steganographic images, which is often limited; and the accuracy of steganographic information extraction, which is frequently inadequate. Inspired by [22], we propose the carrier secret message method, which introduces a carrier component into the process of mapping secret message to secret noise. The properties of the carrier component enable the generation of carrier secret noise, which in turn enhances the capacity of the secret message to generate steganography images. Martin et al. [23] proposed that the Wasserstein distance enables the computation of the full set of all possible joint distributions. This allows for the estimation of the expected distance between samples to the distribution of the dataset, even when the data is

unevenly distributed. Inspired by [24], we use the Wasserstein distance is integrated into the loss function of the discriminator. This enables us to assess the joint distribution between the cover image and the generated image, thereby enabling the generator to balance the quality of the steganographic image and the steganographic capacity of the secret message. Furthermore, inspired by [25], we believe that it is difficult for an unconstrained discriminator to balance the relationship between the quality of the generated image and the accuracy of secret message extraction. In order to address this issue, we use weight clipping, which enables the generated image to achieve a balance between the quality of the generated image and the accuracy of secret message extraction.

In summary, this paper proposes image steganography without embedding by carrier secret message for secure communication in networks. This method introduces the Wasserstein distance [26] as the composition of the loss function to the discriminator in the adversarial generation network. The enhanced discriminator for the training effect of the generator, the generator can better produce the steganography image by the steganography image by the secret message noise tensor mapped by the steganography information. At the same time, the carrier wave is introduced as a way to generate the noise tensor of secret message. The carrier wave method can increase the distance between each secret message and restore the distribution of noise so that the generated image will not lead to the appearance of unity. In this paper, the proposed method can effectively solve this problem that the payload of the steganography deep learning technique is not high, and can effectively and stably extract the steganographic information. This paper's contributions are summarized as:

1. We study the way in which carrier loads secret message. This method can effectively improve the accuracy of secret message extraction while ensuring a certain image quality.

2. Based on the method of weight clipping, we design a restricted discriminator to identify the refined part of the image details, which can effectively improve the steganographic capacity of secret message while ensuring the quality of the image.

3. Based on the Wasserstein distance, we designed the loss function of the discriminator, which can adjust the image quality and the accuracy of secret message extraction, and conducted comparative experiments on the subsets of the LSUN dataset Bedrooms, Churches, and FFHQ.

The remaining paper is organized as follows: In the second section, the related work of this paper is introduced, which mainly includes the traditional image steganography and the deep learning based image steganography. The third section describes the methods and experimental procedures thoroughly, the experiment and analysis are presented in the fourth section, and the fifth section summarizes the proposed methods and outlook on prospective work.

## 2 Related work

With the increasing awareness of network user security, secure network communication [27, 28] of message has received widespread attention. Image steganography has received widespread attention from scholars due to its characteristics. At the present time, two main types of image steganography [29]: traditional image steganography and deep learning [30, 31] based image steganography. Image steganography is able to hide secret message in cover images. It is a widely employed method in the fields of message security transmission [32–34], privacy protection [35], and copyright protection [36].

## 2.1 Traditional image steganography

In the field of traditional steganography, it can be classified into two main categories based on the embedding region: spatial domain steganography and frequency domain steganography.

Ma et al. [37] proposed a hierarchical embedding RDHEI method that is capable of generating hierarchical labels from plaintext image distributions. This method categorizes hierarchical labels into three distinct categories: small, medium, and large. The labels are then used to categorize the types of plaintext images. Through predictive technology, hierarchical label maps are calculated before image steganographic embedding can increase the payload capacity while ensuring full reversibility of data. The secret information is precompressed and embedded into the carrier image. In comparison to previous spatial domain technologies, the RDHEI method of hierarchical embedding exhibits a higher load capacity. However, the method employs image steganography in the spatial domain, which will result in the loss of secret information in the steganography image if robustness attacks are performed on the steganography image.

The traditional image steganography of frequency domain transformation hides the secret data within the frequency domain of the cover image, which can significantly decrease the drop of secret data caused by robustness attacks. Giboulot Q et al [38] propose JPEG-based steganography and quantization table modification, by segmenting the carrier image into 8x8 pixel non-overlapping blocks and converting each non-overlapping into DCT coefficient by DCT respectively and designing an encryption algorithm for information encryption to become secret information. Conversion of spatial domain into frequency domain into 2D cosine wave. The secret information is then concealed within the DCT ratio, according to a pre-designed algorithm. However, the method generates steganography images that are susceptible to being identified by a steganography analysis tool when the method is subjected to a deep neural network steganography analysis tool that has been trained against it will leading to insecure transmission of secret information.

Mandal P C et al. [39] proposed an integer wavelet transform (IWT)-based steganography that uses the LSB method and coefficient value difference to utilise wavelet coefficients in approximate and diagonal subbands and in horizontal and vertical subbands, respectively. Since the perceptual range of the low-frequency coefficient is highly sensitive to the human eye, in this method, a small amount of secret information is embedded in the low-frequency coefficient to reduce the imperceptibility of the hidden image to the human eye after the image is hidden, and the threshold is set to ensure that the generated steganographic image has better visual quality. Nevertheless, the quality of the steganography image is guaranteed by the threshold setting method employed in the method, which results in the overall scheme being overly reliant on the expertise of the experts, thereby reducing its versatility.

## 2.2 Image steganography based on generation network

Goodfellow et al. [20] proposed a method for generating confrontation network GANs, which utilise a generator network and a discriminator network to combat the generation of images. The generator generates the image by inputting random noise, while the discriminator identifies the authenticity of the generated image. Both are trained against each other in order to improve the quality of the generated image. Furthermore, Vijay et al. [40] explored the current state of development of image steganography based on deep learning, which provides a viable direction for subsequent researchers.

Yu et al. [41] proposed that the combination of GAN and image steganography in SGAN verifies the advantages of GAN over traditional image steganography and further improves the embedding ability, undetectability and robustness of steganographic images. SGAN generate images through generators, utilising secret information as input to generators in generative

adversarial networks. The discriminator enhances the image produced and the ability of the discriminator to discriminate the input produced image and the original image. The extractor is introduced to extract and recover the secret information in the produced image. However, this scheme only explores the application of GAN in image steganography, which requires specific information data hiding is more challenging and the network model needs to be more carefully tuned to adapt to the secret information capacity of steganography and the quality of the generated images.

Arjovsky et al. [25] proposed the use of Wasserstein distances as a loss function in GANs, with measure the distance between the generated image and the cover image distribution. In contrast to the KL divergence and JS divergence, the Wasserstein distance can still reflect the distance relationship between two regions in the absence of overlapping regions or rarely overlap. The scheme explores the application of Wasserstein distance in GANs network and achieves promising results, but the small capacity of secret information of steganographic images generated by the scheme may lead to incomplete transmission of secret information under high-capacity secret information steganography conditions.

Liu et al. [42] proposed IDEAS, which extracts structural information and texture information from images through the decoupling of the image components. This method uses different structural information and texture information as the input of the training generator to generate steganographic images. Extensive experimentation has proven that conducted to demonstrate that the quality of secret images generated by decoupling information, the accuracy of extracted information and the quality of generated images are significantly enhanced in comparison to previous methods of SWE. Nevertheless, the scheme remains inadequate in its treatment of the relationship between the quality of steganography images and the accuracy of secret information extraction at higher capacity.

The preceding discussion has the limitations of the steganography without embedding. This paper proposes a method of steganography without embedding image steganography that addresses these shortcomings. The method enhances the capability of steganography to analyse secret information in images and the accuracy of extracting that information, while maintaining a certain image quality.

## 3 Methods

This section will introduce our proposed method by carrier secret information for steganography without embedding images. Details of the proposed carrier noise methodology and modelling details are presented, and the experimental structure is shown in Fig 2.

The carrier secret tensor $Z'$ in Fig 2 refers to the carrier secret tensor generated by the secret message through the carrier mapping method. Generator $G_{stru}$ is a generator used to extract structural information. The structure information $S_1$ and $S_2$ refers to a feature tensor representing the structure of an image. Texture information $T_1$ and $T_2$ refers to a feature tensor representing texture in an image. The extractor refers to the ability to extract corresponding information based on the input image or tensor.

### 3.1 Carrier secret tensor generation and extraction

In this paper, we propose a framework for image steganography without embedding, in which secret message is hidden in generated images by carrier secret tensor. The carrier secret tensor can be effectively and accurately extracted from the generated image by an extractor.

If the mapping interval in the input mapping noise of the network is too large, although the accuracy of message extraction can be improved. Nevertheless, the singularity of noise can affect the diversity of generated images, which can result in a high degree of image overlap.

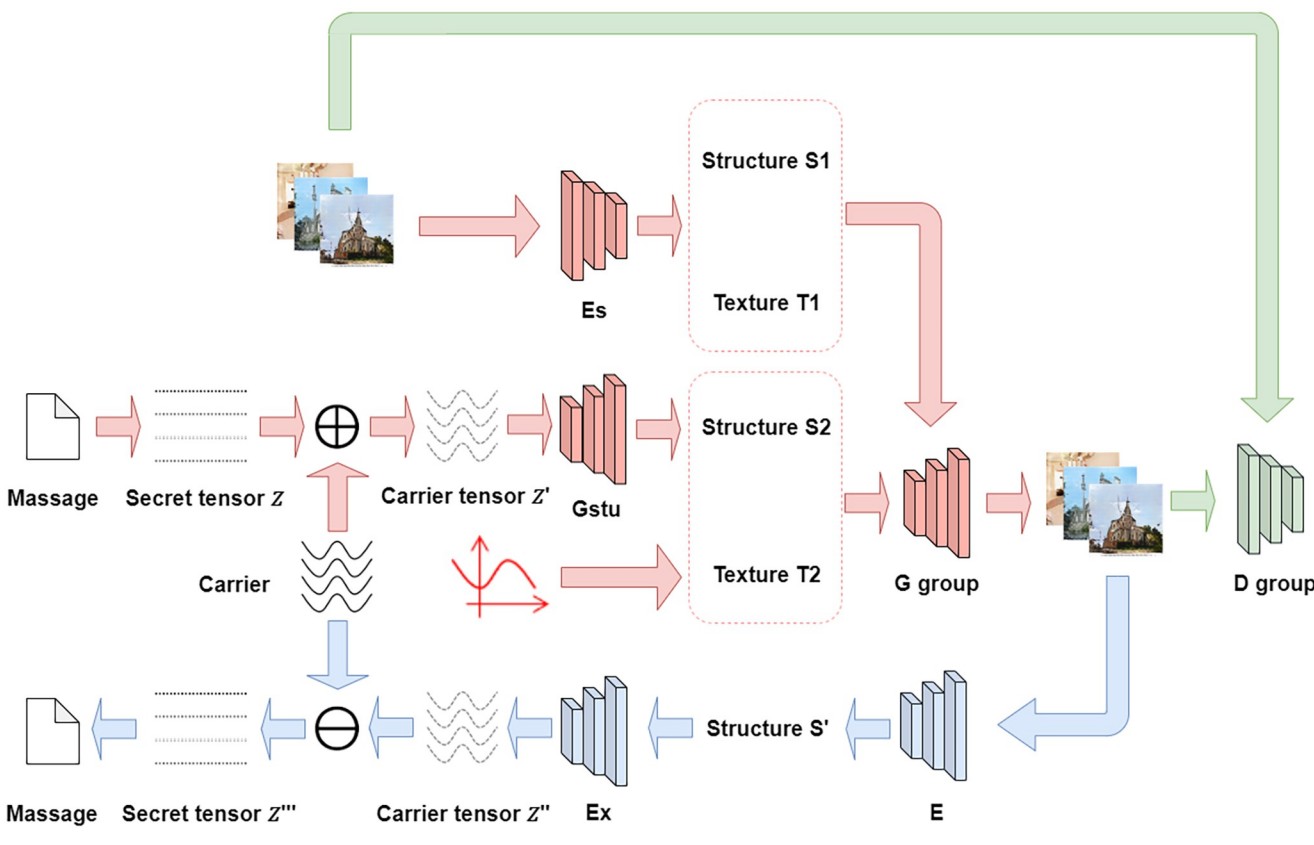

**Fig 2.**

Based on the above problems, the secret noise loaded on the carrier proposed by us can effectively solve the problem and enhance the accuracy of the message extraction process. Simultaneously, it can effectively avoid the situation that the generated image is too single due to the distribution of noise. The specific algorithms and flowcharts for the generation and extraction of carrier noise are provided in Algorithm 1 and Algorithm 2.

**3.1.1 Carrier secret tensor generation process.** The current noise generation process in the SWE technique is to map the secret message $M$ to be transmitted in the noise interval $U$ through the mapping function, and the range of $U$ is $(-1, 1)$, to generate secret noise corresponding to the secret information. The secret message is mapped into $\sigma$-bits noise distribution segments for the mapping function. The overall process is shown in Fig 3. Then, map the decimal value $M$ corresponding to each segment to noise $Z$, and add disturbance in the corresponding interval by adding random noise, the method can be expressed as:

$$Z = \frac{M + 0.5}{2^{\sigma - 1}} - 1 + rand(-\Delta \times r, \Delta \times r) \qquad (1)$$

**Algorithm 1** Carrier Noise Generation Algorithm

**Input:** Secret message $M$, Carrier noise $f(t_m)$, Cover Image $I_{co}$, Extractor $E_s$, Structure generator $G_{stu}$, Stego image generator $G$
**Output:** Stego Image $I_{st}$
1: The secret message $M$ is mapped to the secret tensor $Z$ according to Eq (1)

2: The secret tensor $Z$ is generated according to Eq (2) and carrier
   noise $f(t_m)$ to generate the carrier secret tensor $Z'$
3: Extractor $E_s$ obtains texture information $T_1$ and structural informa-
   tion $S_1$ from the cover image from $I_{co}$
4: The structural generator $G_{stu}$ generates structural information $S_2$
   through the secret tensor $Z'$
5: Obtaining texture information $T_2$ through uniform distribution
   sampling
6: Input $G(T_1, T_2, S_1, S_2)$ generates steganography images $I_{st}$
7: **return** $I_{st}$

In this paper, the carrier secret tensor generation method we propose also needs to generate the transmitted secret message $M$ in the noise interval of $U = (-1, 1)$ through the mapping function, but the noise mapping function divides the mapping interval, the noise $Z$ mapped out from the secret information $M$ is partitioned. Such partition processing will reduce the diversity of the noise, but while reducing the diversity, we load the mapped noise $Z$ on the carrier function $f(t_m)$ The carrier secret tensor $Z'$ is generated on the above, and the diversity of the noise is restored. Its method can be expressed as:

$$Z' = Z + f(t_m) \tag{2}$$

In this paper, the carrier function $f(t_m)$ is selected as $\left(\frac{1+e}{2}\right)^{\sigma-1} \times sin\left(\frac{t_m}{2}\right)$. Here, $t_m$ represents the carrier function input that changes with the length of $m$. It is recommended that the carrier function be chosen within the range $U_f = (-1, 1)$. In this way, the diversity of the noise can be recovered and the secret message can be guaranteed to occupy a dominant position in the noise, to achieve the balance between the carrier function and the secret noise. The process of

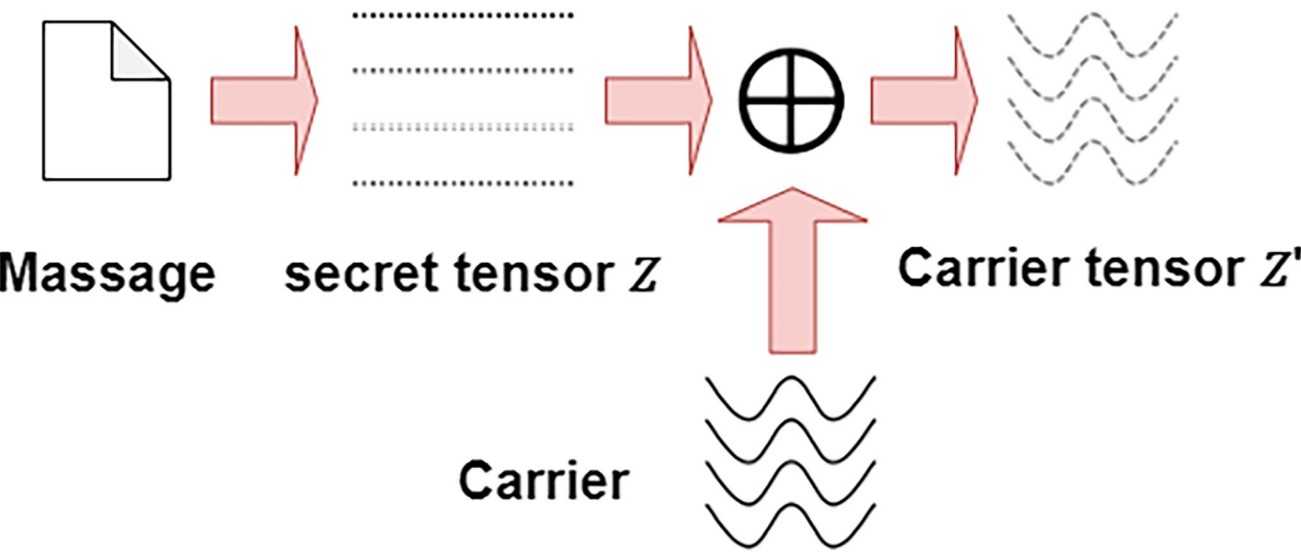

**Fig 3.**

loading the carrier function in the secret message can be expressed as:

$$Z' = f(t_m) + \frac{M + 0.5}{2^{\sigma-1}} - 1$$
$$+ rand\left(-\Delta \times \left(\frac{e}{2} - 1\right)^{\sigma-1}, \Delta \times \left(\frac{e}{2} - 1\right)^{\sigma-1}\right) \tag{3}$$

The loading of carrier noise into the secret noise serves to restore the diversity of the secret noise, thereby enhancing the extraction accuracy of the secret message. The carrier noise generates the structural information $S_2$, which contains the noise $Z'$, through the generator $G_{stu}$. The structural information, in conjunction with the texture information generates the steganography image $I_{st}$ through the image generator $G$. The process can be expressed as follows:

$$S_2 = G_{stu}(Z') \tag{4}$$

$$I_{st} = G(S_1, S_2, T_1, T_2) \tag{5}$$

The discriminator $D_{real}$ is used as the discriminator network to generate steganographic images against the generator $G$ so that the steganographic images are continuously optimized and more difficult to be recognized by human eyes. We will introduce them in 3.3.

**3.1.2 Carrier secret tensor extraction process.** In the part of extracting secret message $M'$, the structural information extractor $E$ is responsible for the extraction of the $I_{st}$ generated by the generator $G$, which in turn generates the structural information $S'$ generated by the carrier noise. The structural information $S'$ is then extracted by the secret tensor extractor $E_x$ to obtain the carrier secret tensor $Z''$. The extraction process of the carrier secret tensor is shown in Fig 4. The process can be expressed as:

$$S' = E(I_{st}) \tag{6}$$

$$Z'' = E_x(S') \tag{7}$$

**Algorithm 2** Carrier Noise Extraction Algorithm

**Input:** Stego image $I_{st}$, Carrier noise $f(t_m)$, Structural information extractor $E$, Extractor $E_x$
**Output:** Extraction secret message $M'$
1: Extraction of structural information $S'$ in $I_{st}$ by structural information extractor $E$ according to Eq (6)
2: The extracted structural information $S'$ by the extractor $E_x$ to extract the carrier secret tensor $Z''$
3: The $Z''$ is obtained by eliminating the $f(t_m)$ according to Eq (8) to obtain the extracted secret tensor $Z'''$
4: The extracted secret tensor $Z'''$ is obtained by inverse mapping the extracted secret meassge $M'$ according to Eq (9)
5: **return** $M'$

The extracted carrier secret tensor $Z''$ is used to extract message through the inverse mapping function. In the inverse mapping function of the extracted message, because the interval is increased during the process of mapping the secret message into the secret tensor, it can be judged by expanding the extraction range, there by reducing the distortion of the secret tensor during image generation. The specific extraction process is to restore the secret tensor $Z'''$ through inverse carrier elimination and extract secret message $M'$ through the inverse mapping function.

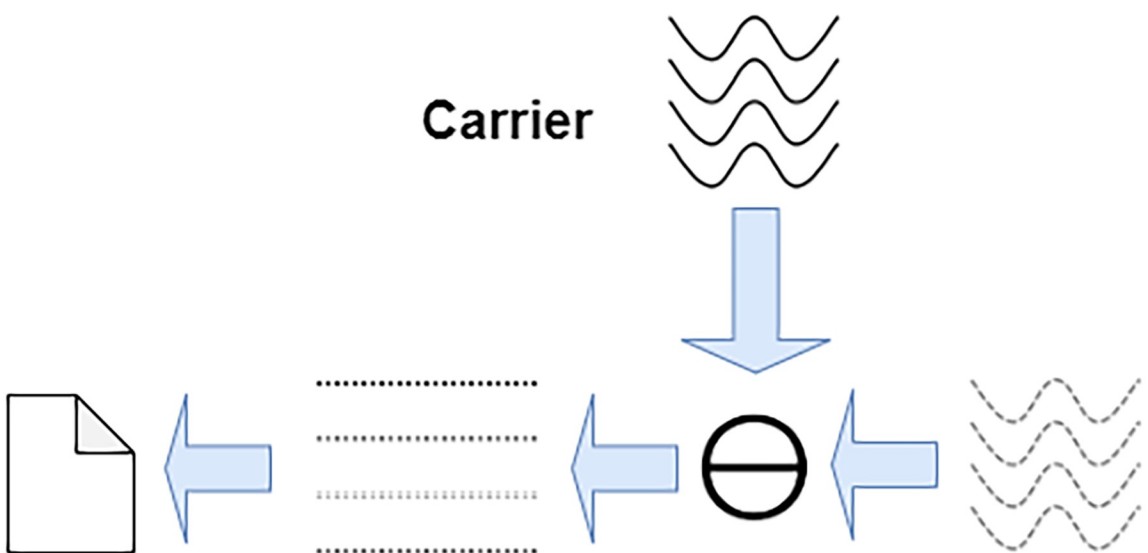

**Fig 4.**

The inverse carrier cancellation process is expressed as:

$$Z''' = Z'' - f(t_m) \tag{8}$$

The inverse mapping function process is expressed as:

$$M' = floor[(Z''' + 1) \times 2^{\sigma - 1}] \tag{9}$$

The extraction of secret message through the secret tensor $Z'''$ involves the removal of the carrier component and the utilisation of the asymmetric range extraction method for the remaining secret message tensor. This method has the capacity to enhance the accuracy of the extracted message, thereby increasing the efficiency and capacity of the transmission of secret message.

## 3.2 Weight clipping

In this paper, the GANs network is employed to generate the steganographic image. GANs networks generate images by means of a confrontation between generators and discriminators. The process is expressed as:

$$min_G max_D V(G, D) = E_{x \sim pdata(x)}[log D(x)] + E_{x \sim p_z(z)}[\log(1 - D(G(z)))] \tag{10}$$

In order to improve the accuracy of extracting the secret message in the generated image, we have limited the weight of the discriminator. In the process, we think that if some weights of the discriminator $D_{real}$ are too large, it will cause the generator to pass the secret message. During the image generation process, instead of the matching rate between the secret message and the generated image in the secret message image, more attention is paid to the distribution of the generated image to approximate the cover image. Although this can come close to the

training data distribution, it can't enhance the generated image quality. Without embedding images, steganography needs to improve the precision of secret message recovery as much as possible under the condition of satisfying the generated image quality. The weights of some parameters in the discriminator are excessive, resulting in the generated image closely resembling the probability of the training data set. This causes a low degree of matching between the image and the secret message, which may lead to the absence or modification of the secret message being undetected. If the weight of the discriminator is insufficient, although the generator is more attentive to the matching rate of the secret message with the generated image. However, the method is unable to approximate the distribution of the training data set, which precludes the assurance of image quality.

In order to ascertain whether the generated images are representative of the true distribution of the dataset, we utilise the discriminator $D_{real}$. Furthermore, we set the weight shear method for the discriminator $D_{real}$. The quality of the image generated by the generator will also be affected if the discriminator is completely unable to tell the difference between the cover image and the generated image. The discriminator weight bounds ensure that $D_{real}$ weights are within a certain range. It is possible that some of the weights of the discriminator, $D_{real}$, may overfit the cover image data during the training process of the discriminator due to the distribution of the generated images. This may result in the deviation of the discriminator weights from the distribution of the secret message. In order to prevent this, it is necessary to restrict the maximum and minimum values of the discriminator $D_{real}$ weights to be used for fitting the distribution of the secret message.

$$clamp(D_{real}) = \begin{cases} p = c_{max} & p > c_{max} \\ p = p & c_{min} \leqslant p \leqslant c_{max} \\ p = c_{min} & p < c_{min} \end{cases} \tag{11}$$

The parameter $p$ represents the weight of a certain neuron of the $D_{real}$ during the training process, and the parameters $c_{max}$ and $c_{min}$ limit the weight of the $D_{real}$. During the training process, if the weight value falls within the interval range of $(c_{max}, c_{min})$, it is advisable to maintain the weight size $p$ at its current value. However, if the weight size $p$ exceeds this range, it is necessary to assign a value to the weight and to limit the weight $p$ variety.

## 3.3 Wasserstein distance structure

The Wasserstein distance structure uses the Wasserstein distance to measure the relationship between the cover image and the generated image. We calculate the smallest Wasserstein distance, which is as close as possible to the cover image distribution relationship. The cover image and the generated image are distinguished by the discriminator $D_{real}$. The margins of the cover images and generated images are represented by the variables $P_r$ and $P_g$ respectively. The *EM* distance between the two margin distributions is calculated the distribution of the generated image margin to that of the actual image margin. Furthermore, this distance is incorporated into the $D_{real}$ loss function to constrain the range of network weights for the aggregated $D_{real}$ through the introduction of weight clipping method. The formula for *EM* can

be expressed as follows:

$$W(P_r \ , \ P_g) = \gamma \sim \prod(Pr, Pg)inf E(x,y) \sim \gamma[||x-y||] \tag{12}$$

$$G_{min}D_{max} = E_{x \sim pdata(x)}[logD(x)] + E_{z \sim p(z)}[log(1 - D(G(z)))] \tag{13}$$

In order to ascertain the distance of required to transform the cover image distribution into the steganography image distribution, the *EM* distance is utilised as loss function in the image distribution loss component of the $D_{real}$. The minimum distance of the joint distribution can be expressed as:

$$E_{x \sim pdata(x)}[D(x)] - E_{z \sim p(z)}[D(G(z))] \tag{14}$$

The generated image includes $X_1$ generated by the cover image $S_1$ and $T_1$, $X_2$ generated by the cover image $S_1$ and randomly sampled $T_2$, $X_3$ generated by $G_{stu}$ structure generator generates structural information $S_2$ based on secret message and randomly sampled $T_2$.

The $D_{real}$ loss function has three parts. The first part is the joint discriminant distribution of $X_1$, $X_2$ and $X_3$ of the cover image by $D_{real}$. The result is obtained by the softplus activation function and the original images is calculated and the generated images, the difference of the discriminative mean as the loss function $L_l$. As shown in Eq 15. The second part is the discriminator for $D_{real}$, which is the *EM* distance between the original images and the generated images $X_1$ using the loss function $L_{X1}$. As shown in Eq 16. The third part is the discriminator $D_{real}$ for the cover image and the generated image $X_w$, where $X_w$ is selected with the training process. As shown in Eq 17. When using the Discriminator for confrontation training, the training process is divided into an upper half and a lower half. The half part is 80% of the total number of iterations, and the second half is 20% of the total number of iterations. When the training process is the first half, $X_w = X_2$, and when the training process is the second half $X_w = X_3$. The selection of hyperparameters for the number of iterations in the upper and lower sections of this reference is based on the numerical choice of hyperparameters for the number of iterations in IDEAS proposed by Liu et al. [42].

In this paper, the three parts of the discriminator's loss function $L_{total}$ are weighted separately and the weight parameters $\alpha$, $\beta$ are set. The weighted loss function of the joint identification of the cover image of the first part and the $X_1$, $X_2$, and $X_3$ of the generated image $\alpha$. The loss for the joint discrimination of cover and generated images as a distribution fitted to cover images. However, only $X_2$ and $X_3$ in the generated image are images generated by secret tensors. If the extraction of secret message is to be improved, the weight ratio of $X_2$ and $X_3$ needs to be increased. Its expression can be expressed as:

$$L_l = m(\psi(D(Concat(\hat{X}_1, \hat{X}_2, \hat{X}_3)))) - m(\psi(D((X)))) \tag{15}$$

where $\psi(\cdot)$ denotes the *softplus*$(\cdot)$ function, and $m(\cdot)$ denotes the taking the mean of the image.

In consideration of the quality of the generated image, the second part is introduced. The Wasserstein distance is used as the loss function $L_{x1}$. The weight $\beta$ is used to adjust and limit the generation of the image quality, and the secret message can be improved under the condition of a certain generation quality extraction. Its second part can be expressed as follows:

$$L_{X1} = m(\psi(D(\hat{X}_1))) - m(\psi(D((X)))) \tag{16}$$

The third part of the discriminator $D_{real}$ loss function is $L_{Xw}$. $L_{Xw}$ selects $X_w$ according to the training process. When the training process is the upper part, the loss function is used to adjust the generator's fitting of the structural information, making the generated image more

fitting combined with the needs of structural information, the subsequent structural informa-tion extractor $E$ can completely extract structural information from the generated image and the complete structural information can better extract carrier noise. It can be expressed as:

$$L_{Xw} = m(\psi(D(\hat{X}_w))) - m(\psi(D((X)))) \tag{17}$$

After weighting the discriminator $D_{real}$ loss function through the split of the above three parts, it can be expressed as:

$$L_{total} = L_l * \alpha + L_{X1} * \beta + L_{Xw} * (1 - \alpha - \beta) \tag{18}$$

## 3.4 Summary of experimental methods

Our algorithm obtains the $T$ and $S$ of the image by decoupling the image and uses the Wasser-stein distance as the discriminator loss function standard, which can effectively improve the precision of extracting secret message. Furthermore, the extractor can be better prepared to extract the correct secret message by increasing the distance before mapping the relationship between secret message and noise. However, if the distance between the noises is too large, the generator noisy data will be too homogeneous, and the generated images will exhibit a signifi-cant problem of homogeneity. To solve this problem, we propose that the noise mapped to the secret message be loaded onto the frequency carrier in order to recover the diversity of the secret noise.

By increasing the distance from the secret message to the noise mapping, our method can better prepare to extract the correct secret message. However, if the distance between the noises is too large, the noise data of the generator will be too single, and the generated Image diversity is reduced. To solve this problem, we propose that the noise mapped to the secret message is loaded on the frequency carrier to recover the diversity of the secret noise. The net-work can improve the training of hiding the secret message in the image generated and improve the extracted structure for extracting the secret message in the steganography images. Finally, the accuracy of secret message extraction can be effectively improved by decoupling the image to obtain the $T$ and $S$ of the image, and introducing the Wasserstein distance as the loss function of the discriminator.

## 4 Experimental results

### 4.1 Experimental setup

1. **Comparison model:** The comparison models selected in this article include DCGAN Steg, WGAN Steg, and IDEAS models. The CWSteg model proposed in this article is compared with the comparison model under the same steganographic capacity.

2. **Datasets:** Two publicly available datasets were selected for the dataset. One is the LSUN [43] dataset, using the Bedroom and Churches datasets from LSUN. The other is the FFHQ [44] dataset, which is a facial image dataset. Select 70000 images from each dataset as the training set, and the processing method of the dataset is to normalize the images used for training into images with a size of $256 \times 256$ pixels.

3. **Image quality:** Steganographic images may be encountered in the process of human subjec-tive visual perception and transmission in the process of transmitting information through the internet. The steganalysis tool [45] analyzes whether the image is an image with hidden information, and generates an image for transmission. Quality, security, and

imperceptibility are particularly important. In our experiments, we measure the quality of generated images by evaluating the starting distance of the index FID [46].

4. **Security:** Security is proved through the use of two well-known steganalysis tools, including YeNet [47] and XuNet [48]. The steganalysis tool can evaluate the imperceptibility of the images generated by the model proposed in this paper. Only when the generated image is imperceptible to the human eye and imperceptible to steganalysis tools, which cannot differentiate among images containing secret information, can such steganalysis images provide secure and authentic communication of secret information.

## 4.2 Weight clipping range selection experiments

In selecting the size of the weight clipping, we set the hidden data volume with a conditional steganography capacity of 2304bits ($N = 3$, $\sigma = 3$) on the subset Bedroom in the LSUN dataset. Because the steganography capacity of 2304bits is currently medium level of steganography capacity in the field of SWE, it is able to more accurately reflect the scheme's ability to generate steganography image quality and secret message extraction. We then perturbed $\Delta = 25$ to perform weight clipping for range $C$ experiments. The weight clipping ranges selected for comparative experiments are $(-0.1, 0.1)$, $(-1.0, 1.0)$, $(-2.0, 2.0)$, $(-3.0, 3.0)$, $(-4.0, 4.0)$, $(-5.0, 5.0)$ and the unrestricted weight selection method for comparative experiments. In the Table 1, the Euclidean distance measure $p$ is used to represent the accuracy of the secret message versus the quality of the generated image. The results of the experiments are shown in the Table 1:

We experimented to obtain the results presented in Table 1 and plotted the obtained results in Fig 5. The accuracy of secret message extraction can be improved by cutting the weights of the discriminator. If the weight clipping is too small, it will make it difficult for the discriminator $D_{real}$ to distinguish the distribution between the cover image and the generated image, resulting in the quality of the generated image not being able to increase the quality of the generated image through the generator and the discriminator. In the event that the image quality is inadequate and fails to meet the requisite standards for clandestine image transmission, this result in a less secure transmission of steganography images.

If there is no limit to the weight limit of the discriminator $D_{real}$, the quality distribution of the image will exhibit a tendency to overfit to the quality of the cover image. However, the efficiency of correct extraction of secret message is much lower, and it is of paramount importance to prioritize this objective while maintaining a certain degree of fidelity to the underlying image distribution.

The correct extraction rate of secret message. Therefore, in this paper, we select the weight clipping range $C = (-0.1, 0.1)$ to ensure a sufficiently high quality of the generated image and a high extraction accuracy.

**Table 1. Weight range selection experiment result table.**

| C | ACC/softmax(ACC) | FID/1-softmax(FID) | *p* |
|---|---|---|---|
| (-0.1,0.1) | 100%/0.1723 | 61.43 /0.9368 | 0.9525 |
| (-1.0,1.0) | 95.57%/0.1649 | 34.86 /0.9516 | **0.9658** |
| (-2.0,2.0) | 87.98%/0.1528 | 144.62/0.8549 | 0.8684 |
| (-3.0,3.0) | 67.88%/0.1250 | 140.43/0.8609 | 0.8699 |
| (-4.0,4.0) | 72.48%/0.1309 | 215.12/0.7064 | 0.7184 |
| (-5.0,5.0) | 70.57%/0.1284 | 110.01/0.8973 | 0.9064 |
| None | 68.27%/0.1255 | 180.76/0.7918 | 0.8016 |

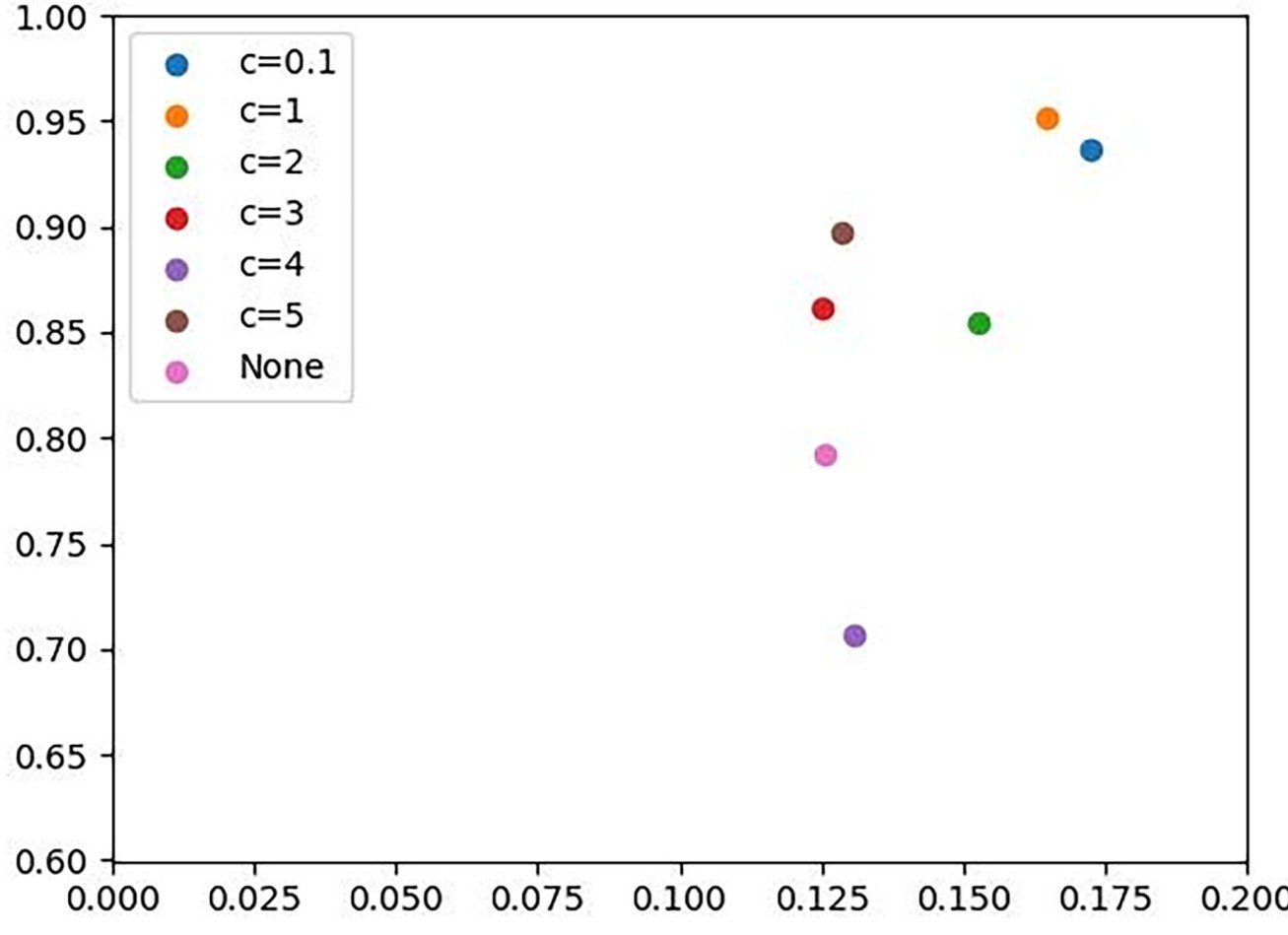

**Fig 5.**

### 4.3 Image quality experiments

In all methods of image steganography, it is necessary to ensure that the human eye cannot distinguish between the cover image and stego image during transmission. Consequently, the quality of the image generated by the method described in this paper must meet this indistinguishability requirement. This subsection presents verification of the quality of steganography images generated by our method. The FID index is the most widely used evaluation index for image generation. It is employed to evaluate the image quality of our proposed method. The experimental results are shown in Table 2:

The our method propose in this paper has similar image quality to the images generated by other methods, which can be concluded from the experimental results in Table 2. Combined with Comparative Experiments section, it can be proved that the proposed method can achieve higher data capacity and secret message extraction accuracy while maintaining image quality.

### 4.4 Anti-detectability performance to the steganalysis

In the without embedding image steganography, in addition to the quality of the generated image needing to meet the indiscernibility of human eyes, the invisibility of steganalysis tools is also important. To solve this problem, we use YeNet and XuNet, which are relatively

**Table 2. Generate image quality experiment result table.**

| Methods | | Bedrooms | Churches | avg.±std.dev. |
|---|---|---|---|---|
| DCGAN-Steg | | 283.32 | 105.79 | 194.55±125.53 |
| SAGAN-Steg | | 159.51 | 99.59 | 129.55±42.37 |
| SSteGAN | | 153.48 | 258.80 | 206.14±74.47 |
| WGAN-Steg | | 147.45 | 181.20 | 164.32±23.86 |
| Ours | | | | |
| N = 1 $\sigma$=1 | $\Delta$=0 | 148.69 | 177.57 | 163.13±20.42 |
| | $\Delta$=25 | 158.06 | 61.09 | 109.57±68.57 |
| | $\Delta$=50 | 95.24 | 103.65 | 99.44±5.95 |
| N = 2 $\sigma$=2 | $\Delta$=0 | 175.10 | 129.95 | 152.52±31.92 |
| | $\Delta$=25 | 203.59 | 46.96 | 125.27±110.75 |
| | $\Delta$=50 | 158.63 | 105.48 | 132.06±37.58 |
| N = 3 $\sigma$=3 | $\Delta$=0 | 175.90 | 81.40 | 128.65±66.82 |
| | $\Delta$=25 | 68.73 | 46.96 | 57.84±15.39 |
| | $\Delta$=50 | 183.19 | 73.83 | 128.51±77.33 |

advanced steganalysis tools, to evaluate the undetectability of generated images for our propose method and other similar methods.

Table 3 shows that the steganographic images generated by our method and the images generated by other benchmark methods are all around 0.5 for steganalysis tools, which means that for steganalysis tools, only cover images and generated images can be randomly judged. The authenticity of the image cannot be effectively distinguished. The results demonstrate that our method is resilient to detection by steganalysis tools and that it is challenging for such tools to discern whether the image generated by the SWE method is genuine. The transfer of secret message can be discovered by steganalysis tools, thus providing a reliable method for the transfer of secret message.

## 4.5 Model steganography capacity performance experiments

The without embedding image steganography algorithm we proposed in this paper has a capacity of Size × N × $\sigma$ for steganographic data. In order to test the accuracy of the steganographic data extraction of our proposed method under different conditions. We conducted the experiments in Table 4 under the conditions of different steganography capacities, different random noise $\Delta$, and different data sets, and verified the correct data extraction rate of our steganography algorithm. Different steganography capacity is mainly to modify the value of $N$ and $\sigma$ to change the steganographic capacity of the image. Furthermore, the quality of images generated by steganography without embedding was enhanced by adjusting the $\Delta$, and experiments were conducted on the LUSN dataset bedroom, Churches experiments with the FFHQ face dataset. Our method effectively prevents detection by steganography image analysis tools

**Table 3. Safety evaluation experiment result table.**

| Methods | DCGAN-Steg | SAGAN-Steg | SSteGAN | WGAN-Steg | Ours | | |
|---|---|---|---|---|---|---|---|
| | | | | | N = 1 $\sigma$=1 | N = 2 $\sigma$=2 | N = 3 $\sigma$=3 |
| Xu-Net | 0.568 | 0.500 | 0.491 | 0.542 | 0.478 | 0.522 | 0.497 |
| Ye-Net | 0.573 | 0.569 | 0.519 | 0.548 | 0.553 | 0.526 | 0.488 |

**Table 4. Experimental results.**

|  |  | Bedrooms | Churches | FFHQ | capacity |
|---|---|---|---|---|---|
| N = 1 $\sigma$=1 | $\Delta$=0 | 100% | 100% | 100% | 256bits |
|  | $\Delta$=25 | 100% | 100% | 100% |  |
|  | $\Delta$=50 | 100% | 100% | 100% |  |
| N = 3 $\sigma$=3 | $\Delta$=0 | 100% | 100% | 100% | 2304bits |
|  | $\Delta$=25 | 100% | 100% | 100% |  |
|  | $\Delta$=50 | 100% | 100% | 100% |  |
| N = 3 $\sigma$=4 | $\Delta$=0 | 100% | 100% | 100% | 3072bits |
|  | $\Delta$=25 | 100% | 100% | 100% |  |
|  | $\Delta$=50 | 100% | 100% | 100% |  |
| N = 4 $\sigma$=4 | $\Delta$=0 | 100% | 99.98% | 100% | 4096bits |
|  | $\Delta$=25 | 100% | 100% | 100% |  |
|  | $\Delta$=50 | 99.93% | 99.98% | 99.98% |  |

during image transmission. This provides a reliable method for the transmission of secret message.

Table 4 and Fig 6 show that our method can ensure high level of quality of generated images and accuracy of secret message extraction. Although the quality of the generated images is degraded, it still has a great advantage over the current SOTA SWE methods at the same capacity.

## 4.6 Comparative experiments

We compare WGAN-Steg, SAGAN-Steg, DCGAN-Steg, SSteGAN and IDEAS with our method under different steganographic capacities. The steganography capacity of the model includes 100bits, 200bits, 256bits ($N = 1$, $\sigma = 1$), 2304bits ($N = 3$, $\sigma = 3$), 3072bits ($N = 3$, $\sigma = 4$) respectively. In addition to that, we still set up the environment with different $\Delta$ superparameters for implementation, which includes $\Delta$=50, $\Delta$=25, $\Delta$=0.

From the Table 5, we see that compared with other models, our propose model has greatly improved the extraction accuracy of secret message with the same capacity. In SWE methods, it is particularly important to accurately extract the hidden information, and our method is not weaker than the current SWE methods in terms of the quality of the images generated by the secret message. At the same time, under different random perturbations $\Delta$, there is a positive correlation with the image quality and a negative correlation with the extraction accuracy, which makes our method meet the needs of different conditions.

## 4.7 Ablation study of the impact of different methods on extraction accuracy

We propose the innovation of Wasserstein distance as the loss function of the $D_{real}$, limit the weight range of the $D_{real}$, and load secret message on the carrier to improve the accuracy of message extraction. At the same time, we design an ablation experiment to prove our innovation points can achieve the expected effect when used alone, and the experimental results are shown in Tables 6 and 7. It is verified that the carrier secret tensor, Weight clipping, and Wasserstein distance loss function can accurately improve the extraction rate of secret message. We performed ablation experiments with a steganography capacity of 2304bits ($N = 3$, $\sigma = 3$) and $\Delta = 50$ on the training set of Bedrooms, as shown in Table 6. In addition, we performed

**Fig 6.**

ablation experiments with a steganography capacity of 3072bits ($N = 3$, $\sigma = 3$) and $\Delta = 25$ on the training set of Churches, as shown in Table 7.

Through Tables 6 and 7, it is proved that the Wasserstein loss function proposed by us, the weight clipping limit $D_{real}$ discriminator and the carrier loaded secret information generation tensor can improve the accuracy of secret information extraction, and can be more effectively improved when combined with different methods extraction accuracy of secret information. The efficacy of the proposed carrier secret tensor, weight clipping, and Wasserstein distance loss function in enhancing the accuracy of secret message extraction can be demonstrated through the use of Tables 6 and 7. Furthermore, the combination of these methods with different secret message extraction methods can result in enhanced effectiveness.

## 5 Limitation and discussion

The proposed method is designed to enhance the steganography capacity and information extraction accuracy of unembedded images, thereby addressing the limitations of low

**Table 5. Compare the experimental results.**

| | | Bedrooms | Churches | FFHQ | capacity |
|---|---|---|---|---|---|
| DCGAN-Steg | | 94.01% | 94.56% | 96.29% | 100bits |
| SAGAN-Steg | | 96.77% | 95.86% | 97.12% | 200bits |
| SSteGAN | | 98.41% | 97.53% | 97.23% | 100bits |
| WGAN-Steg | | 92.23% | 90.04% | 92.85% | 100bits |
| IDEAS | Δ=0 | 100% | 100% | 100% | 256bits |
| | Δ=25 | 100% | 100% | 100% | |
| | Δ=50 | 99.54% | 99.55% | 99.49% | |
| | Δ=0 | 99.65% | 88.41% | 84.41% | 2304bits |
| | Δ=25 | 86.15% | 99.48% | 83.80% | |
| | Δ=50 | 87.07% | 94.14% | 85.02% | |
| | Δ=0 | 75.78% | 95.05% | 87.11% | 3072bits |
| | Δ=25 | 96.55% | 86.76% | 90.33% | |
| | Δ=50 | 87.50% | 88.12% | 85.42% | |
| Ours | Δ=0 | 100% | 100% | 100% | 256bits |
| | Δ=25 | 100% | 100% | 100% | |
| | Δ=50 | 100% | 100% | 100% | |
| | Δ=0 | 100% | 100% | 100% | 2304bits |
| | Δ=25 | 100% | 100% | 100% | |
| | Δ=50 | 100% | 100% | 100% | |
| | Δ=0 | 100% | 100% | 100% | 3072bits |
| | Δ=25 | 100% | 100% | 100% | |
| | Δ=50 | 100% | 100% | 100% | |

**Table 6. Results of ablation experiments on the Bedroom dataset.**

| Carrier | Clamp | Wasserstein | ACC | Baseline(ACC) |
|---|---|---|---|---|
| ✓ | | | 99.87% | 87.07% |
| | ✓ | | 99.48% | |
| | | ✓ | 99.46% | |
| ✓ | ✓ | | 100% | |
| ✓ | | ✓ | 100% | |
| | ✓ | ✓ | 100% | |
| ✓ | ✓ | ✓ | 100% | |

**Table 7. Results of ablation experiments on the Churches dataset.**

| Carrier | Clamp | Wasserstein | ACC | Baseline(ACC) |
|---|---|---|---|---|
| ✓ | | | 90.29% | 86.76% |
| | ✓ | | 91.48% | |
| | | ✓ | 94.73% | |
| ✓ | ✓ | | 97.06% | |
| ✓ | | ✓ | 96.05% | |
| | ✓ | ✓ | 99.38% | |
| ✓ | ✓ | ✓ | 100% | |

steganography capacity and low extraction accuracy observed in unembedded images. Nevertheless, the scheme is still constrained by other issues. It is evident that the quality of steganography images in our proposed method still requires improvement, particularly in environments where high quality transmission is a necessity. In order to achieve high quality steganography images in low steganography capacity environments, it is necessary to reduce the size of the secret message. This allows for the generation of high quality steganography images for secure message transmission.

## 6 Conclusion

This paper proposes image steganography without embedding on carrier secret information for secure communication in networks. This steganography can effectively enhance the capability of steganography images for secret information hiding and improve the accuracy of secret information extraction. Firstly, within the domain of image steganography without embedding, the high capacity secret message is challenging to guarantee the image quality and the accuracy of the secret message extraction. In order to address this issue, the concept of a carrier secret tensor is introduced. This method ensures that the generated image is guaranteed by mapping secret message into carrier secret tensor. At the same time as ensuring diversity, the accuracy rate of extracting secret message from high capacity secret message is improved. Secondly, introducing the Wasserstein distance as the discriminator's loss function can effectively restrict the discriminator from paying too much attention to the image quality and ignoring the status of the most important secret information in the generated image. Finally, the use of the weight clipping method for the discriminator, the over-adjusted neuron parameters of the discriminator are limited while training, and the balance between the image quality of the steganographic image and the secret information capacity is enhanced and the image quality is maintained at the same time. In conclusion, we propose a piece of carrier-based secret message without embedding image steganography based on the Wasserstein distance. The experimental results demonstrate the capacity to enhance the steganographic capacity for secret data and the accuracy of secret data extraction. Although it effectively improves the accuracy of information extraction, the quality of generated images still needs to be improved. In future work, we plan to further improve the quality of its images while maintaining the secret information capacity and extraction accuracy.

## Supporting information

**S1 File.**
(DOCX)

**S2 File.**
(DOCX)

## Author Contributions

**Conceptualization:** Hui Dou.

**Data curation:** Yangwen Zhang.

**Formal analysis:** Yun Luo.

**Investigation:** Yuling Chen.

**Methodology:** Yangwen Zhang, Yuling Chen.

**Resources:** Yuling Chen.

**Software:** Hui Dou.

**Supervision:** Yuling Chen, Hui Dou, Chaoyue Tan, Yun Luo, Haiwei Sang.

**Validation:** Yangwen Zhang, Haiwei Sang.

**Visualization:** Hui Dou, Chaoyue Tan.

**Writing – original draft:** Yangwen Zhang.

**Writing – review & editing:** Yuling Chen.

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
