## [Decision Letter · Decision Letter 0]

3 Jun 2024

PONE-D-24-16179Image Steganography Without Embedding by Carrier Secret Information for Secure Communication in NetworksPLOS ONE

Dear Dr. Chen,

Thank you for submitting your manuscript to PLOS ONE. After careful consideration, we feel that it has merit but does not fully meet PLOS ONE’s publication criteria as it currently stands. Therefore, we invite you to submit a revised version of the manuscript that addresses the points raised during the review process.

We look forward to receiving your revised manuscript.

Kind regards,

Feng Ding

Academic Editor

PLOS ONE

Journal Requirements:

   "This research was supported by Foundation of National Natural Science Foundation of China (62202118), and Scientific and Technological Research Projects from Guizhou Education Department (Qian jiao ji [2023]003 ), 

and Provincial Department of Science and Technology's Hundred level Innovation Talents Project (Guizhou Science and Technology Cooperation Platform Talents-GCC [2023] 018), Guizhou Province Major Project 

(Qiankehe Major Project No. [2024] 003), and Top Technology Talent Project from Guizhou Education Department (Qian jiao ji [2022]073). (By Corresponding authors: Yuling Chen). Guizhou Provincial Basic Research Program(Natural Science):ZK[2024](652) and Science and Technology Program of GuiYang:(ZK[2024]-1-2). (by Authors: Haiwei Sang)"

5. We note that Figure 1,2 and 6 includes an image of a participant in the study. 

6. Please remove your figures from within your manuscript file, leaving only the individual TIFF/EPS image files, uploaded separately. These will be automatically included in the reviewers’ PDF.

Reviewers' comments:

Reviewer's Responses to Questions

**Comments to the Author**

1. Is the manuscript technically sound, and do the data support the conclusions?

Reviewer #1: Partly

Reviewer #2: Yes

2. Has the statistical analysis been performed appropriately and rigorously? 

Reviewer #1: Yes

Reviewer #2: Yes

3. Have the authors made all data underlying the findings in their manuscript fully available?

Reviewer #1: Yes

Reviewer #2: Yes

4. Is the manuscript presented in an intelligible fashion and written in standard English?

Reviewer #1: Yes

Reviewer #2: Yes

5. Review Comments to the Author

Reviewer #1: 1. The abstract needs substential revision. Please write the abstact as: 2 lines with introducing the work, next 2 lines with presenting the peoblem and objectives, and then 2-3 lines discussing the methods and results.

2. The literature review needs to be strenghtened.

3. The research issue and objective of the work is missing.

4. The recnt and related works should be discussed with its merits and issues: DOI: 10.4018/IJDCF.318666, https://doi.org/10.1007/s12652-021-03365-9, https://doi.org/10.3390/electronics12051222, Digital image steganography techniques in spatial domain: A study, https://doi.org/10.1016/j.jisa.2023.103541

5. Present the embedding and extraction process in an algorithmic way.

6. Provide numerical illustration.

7. Validate the results with proper experimentation and sufficient performance measuring parameters.

8. Mention the limitation and implications.

Reviewer #2: The manuscript presents a novel approach to image steganography without embedding by utilizing carrier secret information for secure communication. The method leverages the Wasserstein distance and weight clipping to enhance the capacity and accuracy of secret data extraction while maintaining image quality. In general, the manuscript is well-structured and also provides a comprehensive experimental evaluation. However, several areas require improvement for clarity and completeness.

1. There are several grammatical errors and awkward phrasings throughout the manuscript. A thorough proofreading is necessary to improve readability. For example, "the covert information can increase the covert information capacity of steganography" could be rephrased for clarity.

2. Ensure that all figures and tables are correctly labeled and referenced in the text. Some figures are referenced in a way that is not immediately clear.

3. The related work section is comprehensive but could benefit from a more critical analysis of the limitations of existing methods. This would better highlight the novelty and advantages of the proposed approach. Besides, the title of Section 2.2 does not align with the content discussed. It is suggested to refine the title to better reflect the scope of the section, e.g., Image Steganography Based on Cover Generation.

4. The introduction should provide a clearer motivation. Why is the proposed method important, and what specific gap does it fill in the current literature?

5. The terminologies should be standardized, e.g., "picture" should be consistently referred to as "image."

6. PLOS authors have the option to publish the peer review history of their article (what does this mean?). If published, this will include your full peer review and any attached files.

Reviewer #1: No

Reviewer #2: No

---

## [Author Response · Author response to Decision Letter 0]

18 Jun 2024

RESPONSE LETTER

Manuscript Number: PONE-D-24-16179

Title: “Image steganography without embedding by carrier secret information for secure communication in networks”

Authors: Yangwen Zhang, Yuling Chen, Hui Dou, Chaoyue Tan, Yun Luo, Haiwei Sang 

Dear editor and reviewers,

Thank you very much for your time involved in reviewing the manuscript. On behalf of all the contributing authors, I would like to express our sincere appreciations of your constructive comments concerning our article entitled “Image Steganography Without Embedding by Carrier Secret Information for Secure Communication in Networks” (Manuscript No.: PONE-D-24-16179). These comments are all valuable and helpful for improving our article. According to your comments, we have made extensive modifications to our manuscript to make our results convincing. 

In addition, we promise to continuously revise our paper until all reviewers and the editor are satisfied. The following are the comments, our responses, and the detailed revisions that we have made. Finally, we would like to thank all of you very much for your constructive comments and positive support for this manuscript.

 

Response to Reviewers:

 Response to Review 1

Comment 1: The abstract needs substential revision. Please write the abstact as: 2 lines with introducing the work, next 2 lines with presenting the peoblem and objectives, and then 2-3 lines discussing the methods and results.

Response 1: Thank you very much for the reviewers' suggestions. In response to your question, we have carefully revised the abstract section of the article. The first 2 lines introduced the work of image without embedding steganography. In the next 2 lines, we will introduce the existing problems of image without embedding steganography and the objectives of this article. In the last 3 lines, we discussed the method proposed in this article and its effectiveness. With the following modifications in the manuscript：

(Abstract) (Page 1)

Steganography, the use of algorithms to embed secret information in a carrier image, is widely used in the field of information transmission, but steganalysis tools built using traditional steganographic algorithms can easily identify them. Steganography without embedding (SWE) can effectively resist detection by steganography analysis tools by mapping noise onto secret information and generating secret images from secret noise. However, most SWE still have problems with the small capacity of steganographic data and the difficulty of extracting the data. Based on the above problems, this paper proposes image steganography without embedding carrier secret information. The objective of this approach is to enhance the capacity of secret information and the accuracy of secret information extraction for the purpose of improving the performance of security network communication. The proposed technique exploits the carrier characteristics to generate the carrier secret tensor, which improves the accuracy of information extraction while ensuring the accuracy of secret information extraction. Furthermore, the Wasserstein distance is employed as a constraint for the discriminator, and weight clipping is introduced to enhance the secret information capacity and extraction accuracy. Experimental results show that the proposed method can improve the data extraction accuracy by 10.03% at the capacity of 2304 bits, which verifies the effectiveness and universality of the method. The research presented here introduces a new intelligent information steganography secure communication model for secure communication in networks, which can improve the information capacity and extraction accuracy of image steganography without embedding.

Comment 2: The literature review needs to be strenghtened.

Response 2: Thank you very much for your suggestions. Based on your suggestion, we have made modifications and enhancements to the literature review section. It mainly enriches the content of the literature review and expresses the advantages and limitations of its work on the content of the literature review. In addition, we also discussed the advantages and limitations of existing work in related work to enhance the novelty and advantages of the proposed methods in the manuscript. The corresponding changes in the manuscript are as follows:

(Lines 2 to 11) (Page 1)

Information hiding techniques include steganography [1, 2] and watermarking [3, 4]. Digital watermarking is the process of embedding a watermark into a cover image by means of an algorithm or a deep neural network, which is used to protect the copyright of an image without the human eye being able to detect the watermark. Aditya [4] proposes a logistic map based fragile watermarking technique, which takes advantage of the sensitivity property of the logistic map to generate the watermark bits. Steganography is the process of hiding secret messages within a medium to create a steganographic carrier, which is then used to secure communication in networks [5], where steganography using images as the carrier medium is known as image steganography.

(Lines 64 to 66 and Lines 84 to 104) (Page 3)

The first strategy is to generate the corresponding hash sequences [17, 18] through a mapping association mechanism that involves the secret message and the set of available images.

The majority of existing SWE methods are afflicted by a number of shortcomings. These include the quality of steganographic images, which is often insufficient to achieve the desired effect of cover images; the capacity of steganographic information in steganographic images, which is often limited; and the accuracy of steganographic information extraction, which is frequently inadequate. Inspired by [22], we propose the carrier secret message method, which introduces a carrier component into the process of mapping secret message to secret noise. The properties of the carrier component enable the generation of carrier secret noise, which in turn enhances the capacity of the secret message to generate steganography images. Martin et al. [23] proposed that the Wasserstein distance enables the computation of the full set of all possible joint distributions. This allows for the estimation of the expected distance between samples to the distribution of the dataset, even when the data is unevenly distributed. Inspired by [24], we use the Wasserstein distance is integrated into the loss function of the discriminator. This enables us to assess the joint distribution between the cover image and the generated image, thereby enabling the generator to balance the quality of the steganographic image and the steganographic capacity of the secret message. Furthermore, inspired by [25], we believe that it is difficult for an unconstrained discriminator to balance the relationship between the quality of the generated image and the accuracy of secret message extraction. In order to address this issue, we use weight clipping, which enables the generated image to achieve a balance between the quality of the generated image and the accuracy of secret message extraction.

(Lines 135 to 182) (Page 4),

With the increasing awareness of network user security, secure network communication [27, 28] of message has received widespread attention. Image steganography has received widespread attention from scholars due to its characteristics. At the present time, two main types of image steganography [29]: traditional image steganography and deep learning [30, 31] based image steganography. Image steganography is able to hide secret message in cover images. It is a widely employed method in the fields of message security transmission [32–34], privacy protection [35], and copyright protection [36].

Ma et al. [37] proposed a hierarchical embedding RDHEI method that is capable of generating hierarchical labels from plaintext image distributions. This method categorizes hierarchical labels into three distinct categories: small, medium, and large. The labels are then used to categorize the types of plaintext images. Through predictive technology, hierarchical label maps are calculated before image steganographic embedding can increase the payload capacity while ensuring full reversibility of data. The secret information is precompressed and embedded into the carrier image. In comparison to previous spatial domain technologies, the RDHEI method of hierarchical embedding exhibits a higher load capacity. However, the method employs image steganography in the spatial domain, which will result in the loss of secret information in the steganography image if robustness attacks are performed on the steganography image.

The traditional image steganography of frequency domain transformation hides the secret data within the frequency domain of the cover image, which can significantly decrease the drop of secret data caused by robustness attacks. Giboulot Q et al [38] propose JPEG-based steganography and quantization table modification, by segmenting the carrier image into 8x8 pixel non-overlapping blocks and converting each non-overlapping into DCT coefficient by DCT respectively and designing an encryption algorithm for information encryption to become secret information. Conversion of spatial domain into frequency domain into 2D cosine wave. The secret information is then concealed within the DCT ratio, according to a pre-designed algorithm. However, the method generates steganography images that are susceptible to being identified by a steganography analysis tool when the method is subjected to a deep neural network steganography analysis tool that has been trained against it will leading to insecure transmission of secret information.

Mandal P C et al. [39] proposed an integer wavelet transform (IWT)-based steganography that uses the LSB method and coefficient value difference to utilize wavelet coefficients in approximate and diagonal subbands and in horizontal and vertical subbands, respectively. Since the perceptual range of the low-frequency coefficient is highly sensitive to the human eye, in this method, a small amount of secret information is embeded in the low-frequency coefficient to reduce the imperceptibility of the hidden image to the human eye after the image is hidden, and the threshold is set to ensure that the generated steganographic image has better visual quality. Nevertheless, the quality of the steganography image is guaranteed by the threshold setting method employed in the method, which results in the overall scheme being overly reliant on the expertise of the experts, thereby reducing its versatility.

3. Luo L, Chen Z, Chen M, Zeng X, Xiong Z. Reversible image watermarking using interpolation technique. IEEE Transactions on information forensics and security. 2009;5(1):187–193.

4. Sahu AK. A logistic map based blind and fragile watermarking for tamper detection and localization in images. Journal of Ambient Intelligence and Humanized Computing. 2022;13(8):3869–3881

17. Zhang X, Peng F, Long M. Robust coverless image steganography based on DCT and LDA topic classification. IEEE Transactions on Multimedia. 2018;20(12):3223–3238.

18. Zou L, Sun J, Gao M, Wan W, Gupta BB. A novel coverless information hiding method based on the average pixel value of the sub-images. Multimedia tools and applications. 2019;78:7965–7980.

22. Ren W, Xu Y, Zhai L, Wang L, Jia J. Fast carrier selection of JPEG steganography appropriate for application. Tsinghua Science and Technology. 2020;25(5):614–624.

23. Arjovsky M, Chintala S, Bottou L. Wasserstein GAN. 2017;.

24. Gulrajani I, Ahmed F, Arjovsky M, Dumoulin V, Courville AC. Improved training of wasserstein gans. Advances in neural information processing systems. 2017;30.

35. Sahu AK, Gutub A. Improving grayscale steganography to protect personal information disclosure within hotel services. Multimedia Tools and Applications. 2022;81(21):30663–30683.

36. Sahu AK, Sahu M, Patro P, Sahu G, Nayak SR. Dual image-based reversible fragile watermarking scheme for tamper detection and localization. Pattern Analysis and Applications. 2023;26(2):571–590.

Comment 3: The research issue and objective of the work is missing.

Response 3: Many thanks to the reviewers for their suggestions, which have greatly enhanced our article.

Based on your feedback, we have added the research questions and objectives of this work in the introduction section. Our work mainly focuses on the research problem of low secret information capacity and poor extraction accuracy in the field of image steganography without embedding. Our work can improve the capacity and extraction accuracy of secret information while ensuring certain security conditions. Specific changes in the manuscript are listed below:

(Lines 12 to 25) (Page 2)

In the field of steganography without embedding, steganography without embedding is able to effectively resist the recognition of steganography analysis tools due to the fact that it does not directly employ the embedding of secret message into the cover image, thus achieving the purpose of securely transmitting information. The field of image steganography without embedding has attracted considerable interest among researchers due to its characteristics. However, the majority of studies on image steganography without embedding have been unable to effectively enhance the capacity and extraction accuracy of secret message while maintaining security. Previous methods have been to have limitations in terms of image distortion and the accuracy of message extraction, particularly when dealing with large capacities of steganographic message. This has led to a reduction in the usability of image steganography without embedding. This paper examines the potential of image steganography without embedding, with the objective of enhancing the capacity and extraction accuracy of secret message while maintaining a certain level of security.

Comment 4: The recent and related works should be discussed with its merits and issues: DOI: 10.4018/IJDCF.318666, https://doi.org/10.1007/s12652-021-03365-9, https://doi.org/10.3390/electronics12051222, Digital image steganography techniques in spatial domain: A study, https://doi.org/10.1016/j.jisa.2023.103541

Response 4: Thank you very much for the problems pointed out by the reviewers, which have greatly contributed to enhancing our article. Based on your feedback, carefully read the work of the above article and discuss its advantages and issues in our manuscript. In addition, we have read the follow-up work of the aforementioned paper, including DOI: https://doi.org/10.1016/j.jksuci.2019.07.004, https://doi.org/10.1007/s11042-022-13015-7, and https://doi.org/10.1007/s10044-022-01104-0, and also discussed the advantages and issues of their work in the manuscript. Specific changes in the manuscript are listed below:

(Lines 2 to 11) (Page 1) 

Information hiding techniques include steganography [1, 2] and watermarking [3, 4]. Digital watermarking is the process of embedding a watermark into a cover image by means of an algorithm or a deep neural network, which is used to protect the copyright of an image without the human eye being able to detect the watermark. Aditya [4] proposes a logistic map based fragile watermarking technique, which takes advantage of the sensitivity property of the logistic map to generate the watermark bits. Steganography is the process of hiding secret messages within a medium to create a steganographic carrier, which is then used to secure communication in networks [5], where steganography using images as the carrier medium is known as image steganography.

(Lines 26 to 45) (Page 2)

Traditional image steganography is mainly based on algorithms for embedding of secret information into the carrier image, which is the main part of image steganography. The embedding method is divided into the spatial domain and frequency domain. Representatives of traditional steganography in the spatial domain are the LSB [6, 7], SUNIWARD [8]. Samar et al. [9] proposed a data hiding method that exploits the low embedding capacity and high variability properties of block-wise histogram shifting, thereby enhancing the robustness and embedding capacity of steganography. However, the spatial domain image steganography method will cause the pixel to change v

---

## [Decision Letter · Decision Letter 1]

16 Jul 2024

Image steganography without embedding by carrier secret information for secure communication in networks

PONE-D-24-16179R1

Dear Dr. Chen,

We’re pleased to inform you that your manuscript has been judged scientifically suitable for publication and will be formally accepted for publication once it meets all outstanding technical requirements.

Kind regards,

Feng Ding

Academic Editor

PLOS ONE

Additional Editor Comments (optional):

Reviewers' comments:

Reviewer's Responses to Questions

**Comments to the Author**

1. If the authors have adequately addressed your comments raised in a previous round of review and you feel that this manuscript is now acceptable for publication, you may indicate that here to bypass the “Comments to the Author” section, enter your conflict of interest statement in the “Confidential to Editor” section, and submit your "Accept" recommendation.

Reviewer #1: All comments have been addressed

Reviewer #2: All comments have been addressed

2. Is the manuscript technically sound, and do the data support the conclusions?

Reviewer #1: Yes

Reviewer #2: Yes

3. Has the statistical analysis been performed appropriately and rigorously? 

Reviewer #1: Yes

Reviewer #2: Yes

4. Have the authors made all data underlying the findings in their manuscript fully available?

Reviewer #1: Yes

Reviewer #2: Yes

5. Is the manuscript presented in an intelligible fashion and written in standard English?

Reviewer #1: Yes

Reviewer #2: Yes

6. Review Comments to the Author

Reviewer #1: The authors have made significant improvements in the revised manuscript. I am now recomending to accept the work.

Reviewer #2: (No Response)

7. PLOS authors have the option to publish the peer review history of their article (what does this mean?). If published, this will include your full peer review and any attached files.

Reviewer #1: No

Reviewer #2: No

---

## [Editor Report · Acceptance letter]

29 Jul 2024

PONE-D-24-16179R1 

PLOS ONE

Dear Dr. Chen, 

I'm pleased to inform you that your manuscript has been deemed suitable for publication in PLOS ONE. Congratulations! Your manuscript is now being handed over to our production team.

Kind regards, 

on behalf of

Dr. Feng Ding 

Academic Editor

PLOS ONE